# A simple assay to quantify mycobacterial lipid antigen-specific T cell receptors in human tissues and blood

**Angela X. Zhou**[1,2], **Thomas J. Scriba**[3], **Cheryl L. Day**[4], **Deanna A. Hagge**[5], **Chetan Seshadri**[1,2]*

**1** Department of Medicine, University of Washington, Seattle, Washington, United States of America, **2** Tuberculosis Research and Training Center, University of Washington, Seattle, Washington, United States of America, **3** South African Tuberculosis Vaccine Initiative, Institute of Infectious Disease and Molecular Medicine and Division of Immunology, Department of Pathology, University of Cape Town, Cape Town, South Africa, **4** Department of Microbiology and Immunology, Emory University School of Medicine, Atlanta, Georgia, United States of America, **5** Mycobacterial Research Laboratories, Anandaban Hospital, Kathmandu, Nepal

* seshadri@u.washington.edu

**Data Availability Statement:** All relevant data are within the manuscript and its Supporting information files.

## Abstract

T cell receptors (TCRs) encode the history of antigenic challenge within an individual and have the potential to serve as molecular markers of infection. In addition to peptide antigens bound to highly polymorphic MHC molecules, T cells have also evolved to recognize bacterial lipids when bound to non-polymorphic CD1 molecules. One such subset, germline-encoded, mycolyl lipid-reactive (GEM) T cells, recognizes mycobacterial cell wall lipids and expresses a conserved TCR-α chain that is shared among genetically unrelated individuals. We developed a quantitative PCR assay to determine expression of the GEM TCR-α nucleotide sequence in human tissues and blood. This assay was validated on plasmids and T cell lines. We tested blood samples from South African subjects with or without tuberculin reactivity or with active tuberculosis disease. We were able to detect GEM TCR-α above the limit of detection in 92% of donors but found no difference in GEM TCR-α expression among the three groups after normalizing for total TCR-α expression. In a cohort of leprosy patients from Nepal, we successfully detected GEM TCR-α in 100% of skin biopsies with histologically confirmed tuberculoid and lepromatous leprosy. Thus, GEM T cells constitute part of the T cell repertoire in the skin. However, GEM TCR-α expression was not different between leprosy patients and control subjects after normalization. Further, these results reveal the feasibility of developing a simple, field deployable molecular diagnostic based on mycobacterial lipid antigen-specific TCR sequences that are readily detectable in human tissues and blood independent of genetic background.

## Author summary

Our T cell receptor repertoires are unique to each individual and encode the history of our antigenic exposures. Because of this property, they can act as markers of infection.

**Funding:** Funding was received from the National Institutes of Health (R01-AI125189 to CS). The funders had no role in study design, data collection and analysis, decision to publish, or preparation of the manuscript.

**Competing interests:** The authors have declared that no competing interests exist.

Due to the diversity of T cell receptors, analyzing an individual's repertoire requires advanced sequencing and machine learning methods. However, there are also subsets of T cells that express conserved T cell receptors shared among genetically unrelated individuals. One such subset, germline-encoded, mycolyl lipid-reactive (GEM) T cells, recognizes mycobacterial cell wall lipids. To determine if we could measure expression of these T cells with a simple assay, we developed a quantitative real-time PCR assay that was validated on plasmids and T cell lines. We then determined whether we could detect the GEM T cell receptor sequence in individuals with or without exposure to *Mycobacterium tuberculosis* or *Mycobacterium leprae*. We found that we could detect the GEM T cell receptor in the blood or skin of most individuals, though expression levels did not distinguish between uninfected or infected states. These data show that simple assays to measure mycobacteria-specific T cell receptor sequences are feasible and further advancements focusing on more disease-specific T cell receptors may allow for development of field-friendly diagnostics.

## Introduction

Mycobacterial diseases such as leprosy and tuberculosis (TB) are major causes of morbidity and mortality across the world [1,2]. A major barrier to eradication of these diseases is the lack of timely and accurate diagnosis. In both diseases, T cells feature prominently in pathophysiology and diagnosis. The Ridley-Jopling classification system for leprosy ranges from polar tuberculoid to polar lepromatous in tandem with the functional dichotomy between T-helper type 1 and type 2 cells [3]. In TB, T cell depletion due to HIV co-infection leads to less cavitary pulmonary disease and more extrapulmonary infections, confirming their importance in localizing the infection [4]. 'Latent TB infection' is defined using a tuberculin skin test or interferon-γ release assay (IGRA), which quantify T cell responses to mycobacterial peptide antigens [5]. Thus, T cell-based diagnosis of mycobacterial diseases is already part of the clinical standard of care but further improvements are needed.

T cells express a heterodimeric T cell receptor (TCR) composed of an α and β chain that is expressed at the cell surface and mediates recognition of specific antigens [6]. Because TCRs require interactions with highly polymorphic major histocompatibility complex (MHC) molecules to recognize peptide antigens, the sequence of a TCR is essentially a 'fingerprint' of antigen exposure. Recent advances in high-throughput sequencing have attempted to leverage this property to develop TCR-based diagnosis of infectious diseases. In one study, deep-sequencing of TCR-β chains in the blood in combination with machine learning was able to identify a minimum set of TCRs that could accurately distinguish healthy subjects with and without seroreactivity to cytomegalovirus [7]. A more recent application of this methodology was able to identify a TCR fingerprint of recent or prior SARS-CoV-2 infection, leading to the first FDA-approved molecular diagnostic using this novel approach [8]. Whether and how such an approach could improve the diagnosis of mycobacterial diseases is unknown. The requirement for expensive deep-sequencing technologies and advanced computational methods of analysis may be a challenge to implement in resource-limited settings.

In addition to foreign peptide antigens, T cells have also evolved to recognize bacterial lipid antigens via the highly conserved CD1 antigen presenting system [9]. A unique feature of CD1-restricted T cells is that they sometimes express T cell receptors whose sequence is conserved across genetically unrelated individuals. For example, some T cells that recognize mycolic acid derivatives, which are abundant in the mycobacterial cell wall, always express TCR-α

chains containing a rearrangement of the TCR-ɑ variable (TRAV)-1-2 and TCR-ɑ joining (TRAJ) 9 genes [10]. These cells are known as germline encoded mycolyl-reactive (GEM) T cells and are detectable at high-frequency in the blood of humans with latent and active TB as well as non-human primates after vaccination with BCG [11,12]. Because the GEM TCR-ɑ sequence is conserved across genetically unrelated individuals, quantitative detection should be possible without the need for deep-sequencing or advanced computational methods.

To test this hypothesis directly, we developed a simple quantitative real-time PCR (qPCR) assay for the GEM TCR-ɑ sequence and validated this assay using plasmids and in vitro derived T cell lines. We used our assay to quantify the level of GEM TCR-ɑ expression in blood from subjects with active or latent TB as well as in skin biopsies from individuals with leprosy. In almost all donors tested, we were able to reliably detect the GEM TCR-ɑ sequence even though levels of expression did not distinguish among the various states of mycobacterial infection and disease. Our data provide a proof-of-concept that simple and field-friendly assays to detect mycobacteria-specific TCR sequences can be developed and applied across genetically diverse populations.

## Methods

### Ethics statement

The study was approved by the Institutional Review Board of the University of Washington, the University of Cape Town, and the Nepal Health Research Council (NHRC). For the South Africa Tuberculosis Cohort, written informed consent was obtained from all adult participants, as well as from the parents and/or legal guardians of the adolescents who participated. In addition, written informed assent was obtained from the adolescents. For the Nepal Leprosy Cohort, both written and oral informed consent were given due to high rates of illiteracy. Individuals that were able to read and write provided a signature, and those unable to read and write provided a thumbprint as proof of consent.

### Quantitative PCR

Total RNA was extracted from peripheral blood mononuclear cells (PBMCs) or T cell populations using RNeasy Mini or Micro Kits (Qiagen, Hilden, Germany). RNA was quantified using a NanoDrop spectrophotometer and complementary DNA (cDNA) was generated using High-Capacity cDNA Reverse Transcription Kit (Thermo Fisher Scientific, Waltham, MA). For dermal biopsy samples, generation of cDNA was previously described [13]. Reaction mixtures were prepared using PowerUp SYBR Green Master Mix (Thermo Fisher Scientific, Waltham, MA) and run on Step One Plus Real-Time PCR System (Applied Biosystems, Waltham, MA). Thermocycler conditions included denaturation at 50˚C for 2 min and 95˚C for 2 min, followed by 40 cycles at 95˚C for 15 s and 60˚C for 1 min. Melt curves were generated by increasing temperatures 0.3˚C/s from 60˚C to 95˚C. The cycle threshold values were calculated using Step One Plus software v2.3 (Applied Biosystems, Waltham, MA).

### Primers

Primers for qPCR were obtained from Integrated DNA Technologies (Coralville, IA), and sequences are as follows: GEM TCR-ɑ: forward (fwd) 5' ttgaaggagctccagatgaaag 3', reverse (rev) 5' ctttggagcaggaacaagacta 3', GEM TCR-β: fwd 5' catctgtgtacttctgtgcca 3', rev 5' aggtcgctgtgtgtttgagc 3', TCR-ɑ constant region (TRAC): fwd 5' gtgacaagtctgtctgcctatt 3', rev 5' aaactgtgctagacatgaggtc 3', TCR-β constant region (TRBC): fwd 5' gccctcaatgactccagatac 3', rev 5' ctgtcaagtccagttctacgg 3'.

## Cloning of GEM TCR-α and -β plasmids

GEM-TCR plasmids used for PCR primer validation were cloned using template-switched PCR as previously described [14]. Briefly, RNA was extracted from glucose monomycolate (GMM)-specific T cell line (G03) and cDNA was synthesized and amplified using SMARTer RACE 5' Kit (Clontech Laboratories, Inc., Mountain View, CA) according to the manufacturer's protocol using primers targeting the TCR-α and -β constant region. The primer sequences are as follows: TCR-α constant region: 5′-gattacgccaagcttgttgctccaggccacagcactgttgctc-3′, TCR-β constant region: 5′-gattacgccaagcttcccattcacccaccagctcagctccacg-3′. The RACE products were resolved through a 1% agarose gel and the NucleoSpin Gel and PCR Clean-up kit was used to extract DNA. Then In-Fusion Cloning of RACE Products was performed according to manufacturer's instructions (Clontech Laboratories, Inc., Mountain View, CA). Plasmid DNA was purified using QIAprep Spin Miniprep Kit (Qiagen, Hilden, Germany) and was submitted for sequencing (Genewiz, South Plainfield, NJ).

## GMM-specific T cell lines, PBMCs and culture media

Generation of the GMM-specific T cell line (G10/CS110) used for assay validation was previously described [15]. Briefly, PBMCs were isolated from a healthy South African adult and incubated with GMM-loaded CD1b tetramer. Tetramer positive T cells were sorted using a FACSAria II (BD Biosciences, San Jose, CA) and expanded in culture. Cell lines were then screened using tetramer staining. Sequencing of TCRs was performed using the Immuno-SEQ assay (Adaptive Biotechnologies, Seattle, WA).

PBMCs from healthy adults for assay validation were isolated using a standard Ficoll method from leukoreduction chambers from Bloodworks Northwest and cryopreserved. For qPCR experiments, cryopreserved vials of cells (at least $2x10^6$ cells per vial) were thawed and rested overnight in RPMI 1640 (Gibco, Waltham, MA) supplemented with 10% fetal calf serum (Hyclone, Logan, UT) before counting and RNA isolation.

## Clinical cohorts

South Africa Tuberculosis Cohort: As previously published, 6363 adolescents in the Worcester region of the Western Cape of South Africa were enrolled in a study to determine the incidence and prevalence of tuberculosis infection and disease [15,16]. Individuals were screened for exposure to *Mycobacterium tuberculosis (M.tb)* using tuberculin skin test and interferon-γ release assay (IGRA) QuantiFERON-TB Gold in-tube tests (Cellestis). PBMCs were then isolated using density centrifugation and cryopreserved. For these experiments, 11 IGRA- and 18 IGRA+ patient samples were selected based on availability of PBMCs. In addition, we selected samples of PBMCs from individuals with active tuberculosis from another cohort [15,17]. Participants in that cohort were recruited from the Cape Town region of South Africa and were ≥18 years of age and seronegative for HIV. All had either positive sputum smear microscopy or positive culture for M. tuberculosis, or both. Diagnosis of pulmonary TB was based on epidemiologic history, signs and symptoms, and roentgenographic findings consistent with TB. PBMCs were isolated from those individuals by density centrifugation with Ficoll-Hypaque (Sigma-Aldrich). Samples from 10 individuals with active tuberculosis were selected for this study based on availability of PBMCs.

Nepal Leprosy Cohort: Dermal biopsies were obtained from patients at Anandaban Hospital in Nepal who were enrolled based on clinical presentation of leprosy, as previously described [13]. A portion of these dermal biopsies underwent fixation, mounting in paraffin, histologic staining and examination by leprosy pathologists at the Schieffelin Institute of Health—Research & Leprosy Centre, Karigiri (Tamil Nadu, India) to determine diagnosis and

leprosy class by Ridley Jopling classification. The remaining sample was used for isolation of RNA and converted to cDNA, which was used in our assay. Based on availability of cDNA, we selected 6 samples from leprosy negative individuals, 12 samples from individuals with lepromatous leprosy (10 lepromatous, 2 borderline lepromatous), and 15 from individuals with tuberculoid leprosy (3 tuberculoid, 12 borderline tuberculoid) for our experiments.

### Data analysis

Relative expression of GEM TCR-α and -β sequences normalized to TCR-α and -β controls were calculated by 2^(-delta delta Ct). Statistical analysis was performed using GraphPad Prism v9 software. Differences between groups were first assessed using Kruskal-Wallis test, then, if significant ($p < 0.05$), pairwise testing by Mann-Whitney was performed.

## Results

### Design of a quantitative PCR assay targeting the GEM-TCR

The GEM TCR consists of a nearly invariant TCR-α chain composed of a rearrangement between the TRAV1-2 and TRAJ9 gene segments flanking a canonical 13 amino acid complementarity-determining region 3 (CDR3) sequence [10]. The TCR-β chain is more variable but still largely restricted to the use of TRBV6-2 genes [10]. These observations governed the design of our quantitative PCR assay. For GEM TCR-α, forward and reverse primers were specifically chosen to bind TRAV1-2 and TRAJ9, respectively. These primers flank the conserved CDR3 region (Fig 1A). Because the GEM TCR-β chain was more variable, we designed reverse primers to bind sequences within the TCR-β constant (TRBC) region, while forward primers were chosen specifically to bind TRBV6-2 (Fig 1A). In order to account for total TCR abundance within blood or tissue samples, we also designed two sets of primers that bind within the conserved constant regions of the TCR-α and -β chains, respectively (Fig 1A).

The GEM TCR-α and GEM TCR-β chains were first isolated and cloned from a GMM-specific T cell line using template-switched PCR. We then performed qPCR on these GEM TCR plasmids, prepared in 10-fold serial dilutions starting from 1000pg. Cycle threshold (Ct) values ranged from 9 to 36 with the limit of detection around Ct values of ~34. (Fig 1B). Melt curves from these experiments revealed a single peak for all samples, with the exception for the lowest concentrations of GEM TCR-β, where a minor secondary melt curve was observed (Fig 1C). These data confirm quantitative detection of the GEM TCR using our primers.

### Quantification of GEM-TCR abundance in a mixed T cell population

We next evaluated whether our assay could quantify GEM TCR abundance in a mixed T cell population. T cells from a GMM-specific T cell line, previously described [15], were mixed with PBMCs from a healthy donor at various dilutions prior to RNA extraction, cDNA synthesis and quantification of the GEM TCR and total TCR. We observed a small decrease in expression of total TCR-α and TCR-β in PBMCs compared to GMM-specific T cells, likely representing the lower abundance of T cells in PBMCs compared to a T cell line (Fig 2A, left). As expected, expression of GEM TCR-α was much lower in PBMCs alone compared to GMM-specific T cells alone, with intermediate Ct values corresponding to the various dilutions (Fig 2A, right). Melt curve analysis revealed a single peak in all samples, except for total PBMC in which a secondary peak was observed (Fig 2B). We observed a similar pattern for GEM TCR-β, though the relative decrease in expression in PBMC samples was less than that observed for the GEM TCR-α sequence (Fig 2A). This is especially evident when comparing total TCR-normalized expression of GEM TCR in the GMM-specific T cell line relative to PBMCs (Fig 2C).

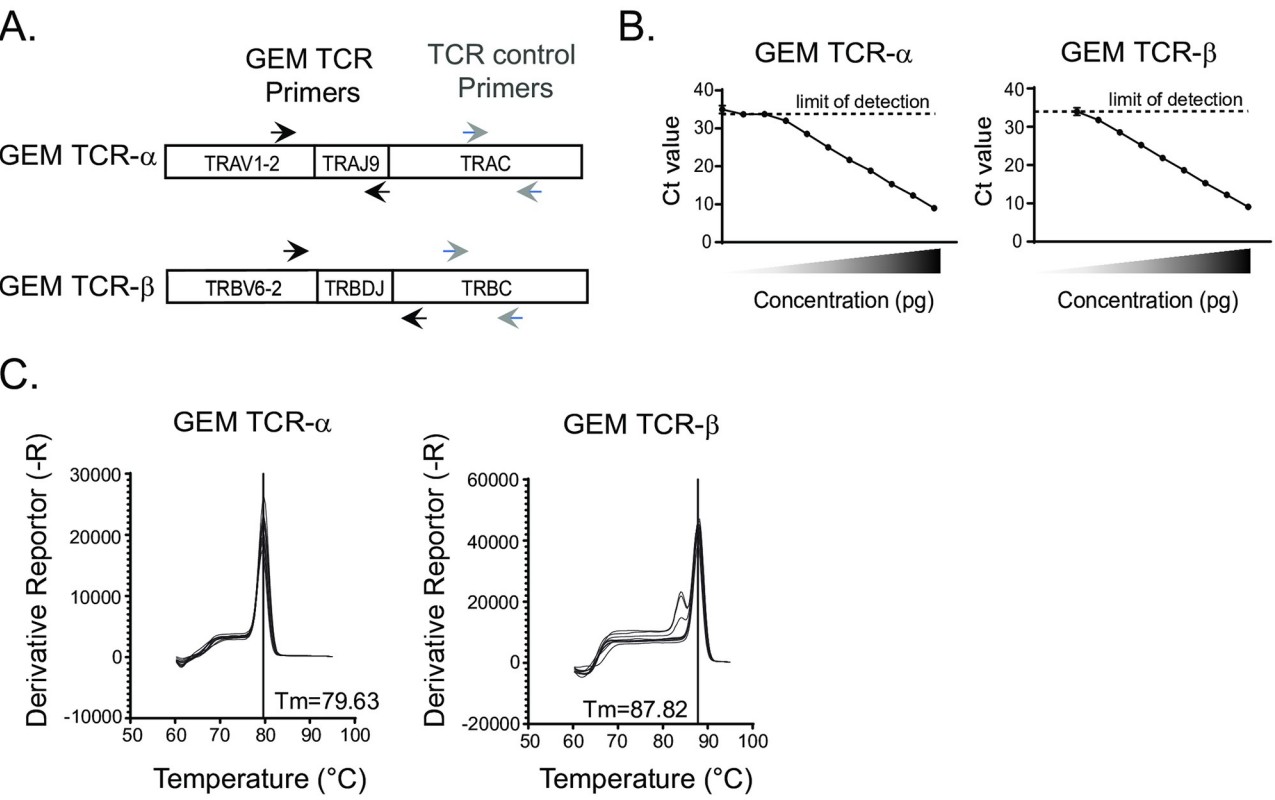

**Fig 1. Design of a quantitative PCR assay targeting the GEM-TCR.** (A) GEM TCR-α primers (forward → and reverse ←) were designed to recognize the TRAV1-2 and TRAJ9 gene segments and flank the CDR3 consensus sequence of the GEM TCR-α. Since GEM TCR-β does not have a conserved CDR3 sequence, primers were designed to specifically recognize TRBV6-2 and the constant region of the TCR-β chain sequence. Control primers for total TCR abundance lie within the TCR-α or TCR-β constant chains (forward → and reverse ←). (B). qPCR primers were validated on plasmids that encode the GEM TCR-α and GEM TCR-β sequences at decreasing concentrations of plasmid. Each dot represents the average cycle threshold (Ct) value of experiments performed in triplicate. (C). Melt curves for GEM TCR-α and GEM TCR-β PCR products with melting temperatures (Tm) of major peak as labeled.

This is likely explained by the lower specificity of the GEM TCR-β primers, leading to multiple PCR products, as demonstrated by the multiple peaks in the melting curves for GEM TCR-β in the samples with greater numbers of PBMCs (Fig 2B). We attempted to validate our assay using genomic DNA (gDNA) instead of cDNA as template but found that the Ct values were too low to support quantitative detection without further modifications to the protocol (S1 Fig). These data demonstrate the capability of our assay to detect differences in abundance of GEM T cells in a mixed population of cells.

## Quantification of GEM-TCR in blood from *M. tuberculosis* exposed subjects

Having validated our assay on T cells, we next evaluated expression of GEM TCR-α in PBMCs isolated from a cohort of individuals in South Africa. This cohort included adults with recently diagnosed active tuberculosis or adolescents who were tested for evidence of prior *M.tb* exposure with both the interferon-γ release assay and tuberculin skin test (S1 Table). In these samples, we were also able to detect expression of GEM TCR-α above the limit of detection in almost all individuals (Fig 3A). There was a slight trend towards higher normalized GEM TCR-α expression in individuals with active TB compared to those without any *M.tb* exposure (p = 0.13) but overall no significant differences between the groups (Fig 3B).

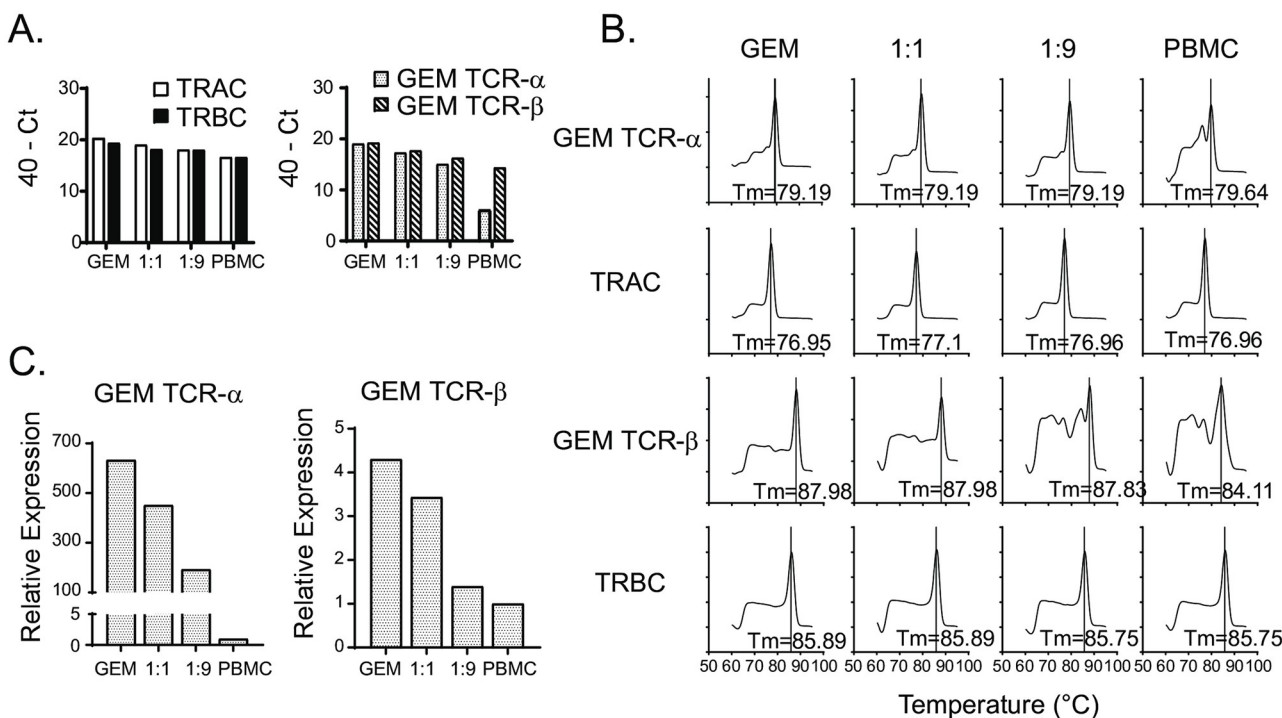

**Fig 2. Quantification of GEM-TCR abundance in a mixed T cell population.** (A). Expression of total TCR (TRAC and TRBC) and GEM TCR-α and -β in a GMM-specific T cell line (GEM) and PBMCs alone, as well as mixed in a 1:1 (GEM:PBMC) and 1:9 ratio. Equal numbers of total cells were used for each condition. (B). Melt curves for each of the qPCR products from the assays in (A). Melting temperatures (Tm) of the major peak is labeled. (C). Relative expression of GEM TCR-α and GEM TCR-β in a GMM-specific T cell line, and mixed GEM:PBMC populations, compared to expression in PBMCs alone. Expression of GEM TCR-α and GEM TCR-β was normalized to total TCR abundance.

## Quantification of GEM-TCR in skin biopsies from leprosy patients

We next wanted to determine whether we could detect GEM TCR in tissues. GMM is known to be produced during infection by *M.tb* and other mycobacterial species [18]. *M. leprae* likely

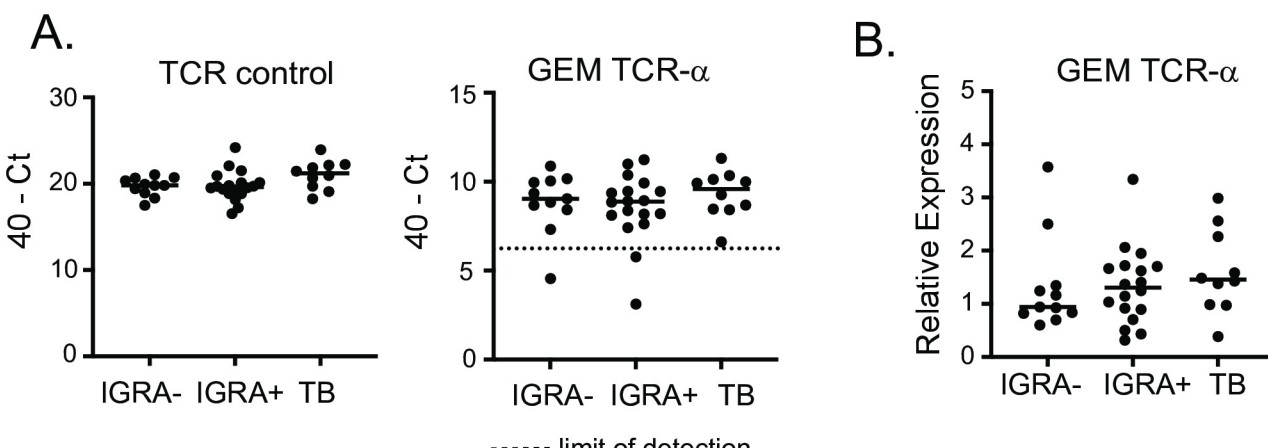

**Fig 3. Quantification of GEM-TCR in blood from *M. tuberculosis* exposed subjects.** PBMCs were isolated from individuals in South Africa with newly diagnosed active TB (TB), with immune reactivity to *M.tb* (IGRA+) or control subjects (IGRA-). qPCR assay was performed on cDNA from these samples. (A). Expression of total TCR-α (left) and GEM TCR-α (right). Limit of detection marked with dashed line (B). Relative expression of GEM TCR-α after normalization to total TCR-α abundance.

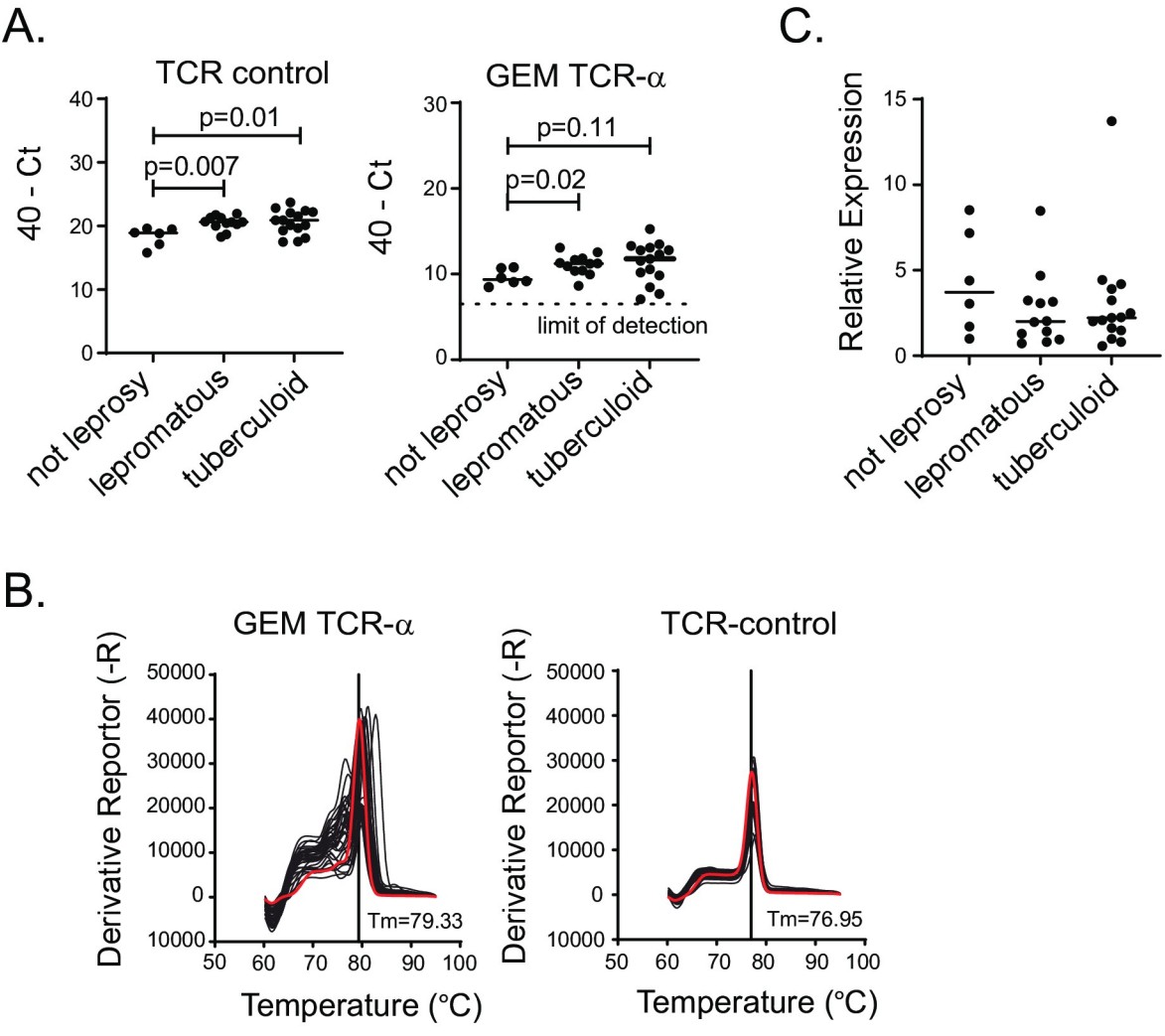

**Fig 4. Quantification of GEM-TCR in skin biopsies from leprosy patients.** Individuals in Nepal with clinical concern for leprosy had dermal biopsy samples collected for histologic examination and RNA isolation, which was converted to cDNA. (A). Expression of total TCR-α (left) and GEM TCR-α (right) in dermal biopsies from individuals with either lepromatous or tuberculoid leprosy compared to controls without leprosy. Each dot represents the average value of duplicates. Limit of detection labeled with dashed line. (B). Melt curves from all samples from leprosy cohort with melting temperature (Tm) of major peak labeled. Red line represents the melt curve from the GMM-specific T cell line. (C). Relative expression of GEM TCR-α after normalization to total TCR-α.

also produces GMM, as the GMM specific T cell clone LDN5 was originally derived from a patient with leprosy [19]. We evaluated expression of GEM TCR in dermal biopsies from Nepalese adults with histologically confirmed lepromatous or tuberculoid leprosy [13]. We also analyzed specimens from control subjects who underwent biopsy but did not receive a diagnosis of leprosy (S2 Table). Biopsy specimens from patients with both tuberculoid and lepromatous leprosy showed higher levels of total TCR-α expression compared to biopsies derived from control subjects (Fig 4A, left). This finding was consistent with recruitment of T cells to the site of infection. In all samples tested, we were able to detect expression of the GEM TCR-α above the limit of detection, and this was higher in lepromatous leprosy patients compared to controls subjects (Fig 4A, right). To confirm that the assay was specifically measuring expression of GEM TCR-α, we examined the melting curves from all samples. We found that there

was a peak around the expected temperature of 79˚C in most samples, suggesting sequences identical to our plasmid and cell line positive controls, but others also contained secondary peaks, particularly around 76˚C (Fig 4B). This suggests possible amplification of non-canonical GEM TCR-α sequences.

After normalizing for total TCR-α, there was no difference in the relative expression of GEM TCR-α between the groups, suggesting that GEM T cells do not undergo preferential recruitment and expansion in leprosy lesions compared to conventional T cells (Fig 4C). These data reveal that GEM T cells constitute part of the skin T cell repertoire and suggest that the recruitment of CD1-restricted T cells to the site of infection may occur independently of Th1 and Th2 cells that typically correlate with histopathological diagnosis [20].

## Discussion

In summary, we developed a simple qPCR assay to measure the abundance of a mycobacterial lipid-specific TCR and used this assay in a cohort of *M.tb*-reactive donors in South Africa and leprosy patients in Nepal. To our knowledge, our results are the first to show that GEM T cells are a natural part of the skin T cell repertoire. In almost all samples tested, we were readily able to detect the sequence above the limit of detection, confirming their designation as 'donor-unrestricted' T cells (DURTs) [21]. Though the abundance of this lipid-specific TCR did not distinguish among healthy controls, infection, or disease, they provide a proof-of-concept that simple, field-friendly assays focused on identifying disease-specific TCRs in tissues and blood are possible.

We focused our efforts on the GEM TCR-α sequence, which has been shown to mediate recognition of mycolic acid and glucose monomycolate [10]. We found that the abundance of GEM TCR-α did not distinguish among mycobacterial disease, infection, or control populations, limiting its use as a diagnostic assay. There are likely several factors that contribute to this. Our overall sample sizes were low, potentially limiting our ability to detect small differences. In addition, though GMM is preferentially produced in the context of infection, both mycolic acid and GMM are present in non-tuberculous mycobacteria and other members of the family Actinomyetales as well as *M. tuberculosis* and *M. leprae* [19,22,23]. Thus, it is possible that GMM-specific T cell responses in control subjects were primed by exposure to environmental mycobacteria. This may be one of the reasons we were able to detect GEM TCR-α sequences in individuals who tested IGRA negative. Next generation assays might focus on lipids with more restricted expression. A natural candidate would be sulfoglycolipids, which are only expressed by virulent mycobacteria [24]. An assay based on sulfoglycolipid-specific TCRs may better differentiate between healthy individuals and those with mycobacterial disease. Currently no equivalent of the GEM TCR-α sequence has been defined for sulfoglycolipids, though patterns in restricted TCR usage are beginning to emerge [25]. Another possibility might be to combine several lipid-specific TCRs into a multiplex assay to improve the specificity of detection.

We found an increased abundance of the GEM TCR-α sequence in dermal biopsies from individuals with lepromatous leprosy compared to control subjects. This increase was correlated with an increase in total TCR-α abundance in both lepromatous and tuberculoid leprosy. There was a trend towards an increase in GEM TCR-α in the skin of individuals with tuberculoid leprosy, though this did not reach statistical significance, possibly due to small sample sizes. These data suggest that in the setting of inflammation, GEM T cells traffic to the site of infection in concert with conventional T cells [26,27]. This is supported by studies that have identified the GEM TCR in TB granulomas in humans [28]. The homing properties of CD1b-restricted T cells are not well-defined. However, CD1a-restricted T cells are known to home to

the skin and have been implicated in various skin conditions [29]. For instance, CD1a-restricted T cells have been shown to express cutaneous lymphocyte antigen and could be isolated from human skin biopsies and produce IL-22 after activation [30,31]. CD1a-restricted T cells from blood and/or skin also recognize antigens implicated in the pathogenesis of skin diseases including psoriasis [32] and atopic dermatitis [33], as well as in wasp and bee venom responses [34,35]. While CD1a is constitutively expressed on epidermal Langerhans cells, CD1b is normally rare in the skin, but has been shown to be upregulated in the dermis after infection with *Borrelia burgdorferi* [36]. Previous studies have also shown increased expression of CD1b proteins in dermal granulomas of individuals with tuberculoid leprosy [37] and human pulmonary TB granulomas [38] raising the potential for recruitment of CD1b-restricted T cells to sites of infection.

Although earlier research suggested that GEM T cells expand in mycobacterial infection [10,15], our data is consistent with recent findings that frequencies of CD1b-GMM or CD1b-MA-specific T cells detected using flow cytometry are not higher in individuals with active TB or a positive IGRA compared to individuals who were IGRA negative [11,39]. Similar results have also been reported for other donor-unrestricted T cells in the context of *M.tb* infection, such as mucosal associated invariant T (MAIT) cells [40,41]. DURTs likely undergo phenotypic changes after pathogen exposure that is not captured by measurements of total cell numbers alone. Further support of this hypothesis comes from a study of individuals challenged with *S*. Paratyphi A who developed enteric fever. Though there was no significant difference in numbers of peripheral blood MAIT cells at one-month post-challenge, there was a significant increase in CD38+ MAIT cells as well as specific clonotypes as defined by TCR-β sequence of sorted cells [42]. We have also recently demonstrated that GMM-specific TCRs that do not conform to the GEM motif are more accurately able to distinguish between TB infection and disease [15]. Thus, incorporation of additional TCRs into the assay may improve its diagnostic accuracy.

The inherent diversity of the TCR repertoire has necessitated the use of high-throughput sequencing, large sample sizes, and machine learning to realize their potential as blood based biomarkers for infectious diseases [43]. Because TB and leprosy are primarily diseases of poverty, it is unlikely that the current approaches will ever find their way into field use for mycobacterial diseases. As an alternative, we show that a simple real-time PCR-based method focused on a single lipid-specific and donor-unrestricted TCR is feasible and yields results above the limit of detection in blood and tissue specimens from all donors tested. While the abundance of GEM-T cells did not differentiate between healthy and disease states, further advancements targeting more disease-specific T cells may allow for the development of simpler, field-friendly diagnostics that would enable greater case detection of patients with TB and leprosy in clinical settings.

## Supporting information

**S1 Fig. Higher sensitivity is achieved using complementary cDNA template compared to genomic gDNA.** Expression of (A) GEM TCR-α and (B) GEM TCR-β from GMM-specific T cell line (GEM) and PBMCs alone, and mixed in a 1:1 (GEM:PBMC) and 1:9 ratio using cDNA template vs gDNA template.
(PDF)

**S1 Table. Demographic data for South Africa Tuberculosis Cohort.** Human samples used in this study are listed by clinical cohort and sample identifier (Sample). The TB cohort (TB) is composed of individuals with pulmonary tuberculosis and either had positive sputum smear microscopy or positive culture for *M. tuberculosis*. Culture was only performed in individuals

who had a negative sputum smear. The adolescent cohort study (ACS) is composed of individuals who were enrolled in a study to determine incidence of tuberculosis infection. Interferon-γ release assay (IGRA) and tuberculin skin test (TST) results are listed and concordantly positive or negative.
(PDF)

**S2 Table. Demographic data for Nepal Leprosy Cohort.** The leprosy cohort is composed of individuals in Nepal who were enrolled based on clinical presentation of leprosy. Bacterial index represents the bacillary load in the biopsy sample. Comprehensive Ridley-Jopling (RJ) indicates how the individual was classified based on comprehensive assessment of all clinical and laboratory data.
(PDF)

**S1 Data. qPCR and melt curve data.** Excel spreadsheet containing, in separate sheets, the underlying numerical data for Figs 1B, 1C, 2A, 2B, 2C, 3A, 3B, 4A, 4B, 4C and S1 Fig.
(XLSX)

## Acknowledgments

Dr. Thomas Hawn for facilitating access to archived cDNA from dermal biopsy specimens. Dr. Krystle Yu for cloning the GEM TCR-α and GEM TCR-β plasmids.

## Author Contributions

**Conceptualization:** Chetan Seshadri.

**Formal analysis:** Angela X. Zhou.

**Funding acquisition:** Chetan Seshadri.

**Investigation:** Angela X. Zhou.

**Resources:** Thomas J. Scriba, Cheryl L. Day, Deanna A. Hagge.

**Supervision:** Chetan Seshadri.

**Writing – original draft:** Angela X. Zhou.

**Writing – review & editing:** Chetan Seshadri.

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
