## [Decision Letter · Decision Letter 0]

18 Oct 2021

Dear Dr. Seshadri,

Thank you very much for submitting your manuscript "A simple assay to quantify mycobacterial lipid antigen-specific T cell receptors in human tissues and blood" for consideration at PLOS Neglected Tropical Diseases. As with all papers reviewed by the journal, your manuscript was reviewed by members of the editorial board and by several independent reviewers. 

We received positive feedback on your manuscript from three of the four reviewers and based on these reviews, we are likely to accept this manuscript for publication, providing that you modify the manuscript according to the review recommendations. 

Sincerely,

Katharina Röltgen

Associate Editor

Richard Phillips

Deputy Editor

Reviewer's Responses to Questions

**Key Review Criteria Required for Acceptance?**

**Methods**

-Are the objectives of the study clearly articulated with a clear testable hypothesis stated?

-Is the study design appropriate to address the stated objectives?

-Is the population clearly described and appropriate for the hypothesis being tested?

-Is the sample size sufficient to ensure adequate power to address the hypothesis being tested?

-Were correct statistical analysis used to support conclusions?

-Are there concerns about ethical or regulatory requirements being met?

Reviewer #1: the description of the patient cohort is sufficient to address the questions raised in the manuscript. The sample sizes are not very high and the selection of samples is based on "availability" rather than prospective or clinically well-defined criteria. However this is considered and discussed adaquately. Ethical approvals were obtained.

Reviewer #2: (No Response)

Reviewer #3: Acceptable

Reviewer #4: (No Response)

**Results**

-Does the analysis presented match the analysis plan?

-Are the results clearly and completely presented?

-Are the figures (Tables, Images) of sufficient quality for clarity?

Reviewer #1: see below

Reviewer #2: These features are okays presented

Reviewer #3: Acceptable

Reviewer #4: (No Response)

**Conclusions**

-Are the conclusions supported by the data presented?

-Are the limitations of analysis clearly described?

-Do the authors discuss how these data can be helpful to advance our understanding of the topic under study?

-Is public health relevance addressed?

Reviewer #1: see below

Reviewer #2: There is no significance of the data presented

Reviewer #3: Acceptable

Reviewer #4: (No Response)

**Editorial and Data Presentation Modifications?**

Reviewer #1: (No Response)

Reviewer #2: The authors have developed a qualitative PCR assay for germline-encoded, mycolyl lipid-reactive (GEM) T cells, recognizes

mycobacterial cell wall lipids, and expresses a conserved TCR-ɑ chain that is shared

among genetically unrelated individuals. This assay was validated on plasmids and T cell lines. and tested on blood samples from South African subjects with or without tuberculin reactivity or with active tuberculosis disease. the authors were able to detect GEM TCR-ɑ above the limit of detection

in 92% of donors but found no difference in GEM TCR-ɑ expression among the three

groups after normalizing for total TCR-ɑ expression (Leprosy, TB diseased vs. healthy/exposed controls). This disappointing results decrease the utility of such a method.

Reviewer #3: Accept

Reviewer #4: (No Response)

**Summary and General Comments**

Reviewer #1: The major novelty of this study is the technical establishment of a molecular assay to measure lipid-specific T cell receptors in clinically relevant samples, including tissue biopsies. The description, presentation and interpretation of the results is very informative and clear. There are obvious limitations of the study: Limited sample size, analysis of two related, but distinct infectious diseases (rather than studying PBMC and biopsies (e.g. lymph nodes) from either tuberculosis or leprosy), no histology from the skin lesions to localize the cells within granulomas and confirm the PCR data). And of course the assay is not relevant for laboratory diagnosis of mycobacterial disease because it does not discrimate between healthy and infected individuals. All these points are critically addressed in the balanced discussion. Tthe authors are careful not to overinterpret the results and rather highlight the technical achievement (see title).

Reviewer #2: (No Response)

Reviewer #3: Zhou et al present an interesting study aimed to develop TCR-based molecular diagnostics of TB and Leprosy. Expanding technologies to timely diagnose neglected diseases like TB and Leprosy is an urgent public health need. The concept of applying TCR-based molecular diagnostics for the detection of infectious diseases has gained attraction with the detection of viral infection using classical MHC-restricted T cell responses against viral antigens. While T cell responses against Mtb immunodominant antigens using interferon-gamma release assays have been successfully applied to measure history of exposure to the bacteria, their utility in diagnosing complex diseases states in Leprosy and TB has been less clear, especially for rare CD1-reactive T cells, like GEM cells. These T cells have the advantage of bypassing HLA polymorphism, making this a logical and valuable direction to pursue. The authors thoroughly validate the diagnostic using plasmids and clones expressing the canonical TCR-alpha found in mycolyl-reactive CD1b-reactive T cell TCR rearrangements and apply it to samples from two cohorts to test this hypothesis. The authors acknowledge the limitation of the assay in distinguishing healthy vs. sick patients in either disease, suggesting limited antigen-specific clonal expansion in disease states. However, the novelty of this approach and rigor in analyzing the relative TCR abundance using valuable clinical cohorts is commendable, and open the concept of using DURTs to develop simple field diagnostics for further study. 

Minor comments:

- It is not clear from the figures if the authors had verified the lower number of N-nucleotides in GEM TRAV1-2-TRAJ-9 rearrangements by sequencing. Was that confirmed?

- In figure 2C, the relative expression of GEM TCR-beta appears to be a hundred-fold lower than GEM TCR-alpha. How do you explain that? I would expect that the GEM TCR-alpha rearrangement to be far less abundant than the GEM TCR-beta counterparts.

- The analysis of lepromatous and tuberculoid leprosy legions is intriguing and important considering that it is ultimately the site of disease in leprosy. Since very little is known about the homing properties of CD1b-reactive T cells, do you hypothesize that CD1b/GMM-reactive T cells would preferentially home to the lesions in a similar manner to CD1a-reactive T cells in autoinflammatory conditions like dermatitis? Please comment in the discussion.

Reviewer #4: 1. Findings looks intersecting and novel approach for the current sittings, but still few things need to be consider from the current approach

2. Did authors able to define a sensitivity and specificity for the developed assays

3. Why do authors think even in the IGRA negative samples GEM-TCRa able to detect more than the limit of detection

4. Authors can also discuss few points about the rationale for this study

5. Please add a detailed demographic profile of the study population as a table in the manuscript.

6. Did all the active TB patients are pulmonary TB cases?

7. Since the samples size per group is very low it has to be mentioned in the study limitations 

8. Please do add AFB smear grade status and culture results status in the demographic tables

9. Authors should discuss few points how this results can be used in the clinical settings or patient management.

PLOS authors have the option to publish the peer review history of their article (what does this mean?). If published, this will include your full peer review and any attached files.

Reviewer #1: No

Reviewer #2: No

Reviewer #3: Yes: Sara Suliman

Reviewer #4: No

Figure Files:

Data Requirements:

Reproducibility:

References

---

## [Editor Report · Decision Letter 1]

23 Nov 2021

Dear Dr. Seshadri,

We are pleased to inform you that your manuscript 'A simple assay to quantify mycobacterial lipid antigen-specific T cell receptors in human tissues and blood' has been provisionally accepted for publication in PLOS Neglected Tropical Diseases.

Thank you for your very thorough response and revision of the manuscript based on the Reviewer's comments and for informing us about the reclassification of one of the samples.

Best regards,

Katharina Röltgen

Associate Editor

Richard Phillips

Deputy Editor

<style type="text/css">p.p1 {margin: 0.0px 0.0px 0.0px 0.0px; line-height: 16.0px; font: 14.0px Arial; color: #323333; -webkit-text-stroke: #323333}span.s1 {font-kerning: none

</style>

---

## [Editor Report · Acceptance letter]

10 Dec 2021

Dear Dr. Seshadri,

We are delighted to inform you that your manuscript, "A simple assay to quantify mycobacterial lipid antigen-specific T cell receptors in human tissues and blood," has been formally accepted for publication in PLOS Neglected Tropical Diseases.

Best regards,

Shaden Kamhawi

co-Editor-in-Chief

Paul Brindley

co-Editor-in-Chief
